# The Effect of Positive Charge Distribution on the Cryoprotective Activity of Dehydrins

**DOI:** 10.3390/biom12101510

**Published:** 2022-10-19

**Authors:** Margaret A. Smith, Steffen P. Graether

**Affiliations:** 1Department of Molecular and Cellular Biology, University of Guelph, Guelph, ON N1G 2W1, Canada; 2Graduate Program in Bioinformatics, University of Guelph, Guelph, ON N1G 2W1, Canada

**Keywords:** intrinsically disordered proteins, cryoprotection, dehydrins, lactate dehydrogenase, yeast frataxin homolog 1, charge, sequence order, circular dichroism

## Abstract

Dehydrins are intrinsically disordered proteins expressed ubiquitously throughout the plant kingdom in response to desiccation. Dehydrins have been found to have a cryoprotective effect on lactate dehydrogenase (LDH) in vitro, which is in large part influenced by their hydrodynamic radius rather than the order of the amino acids within the sequence (alternatively, this may be a sequence specific effect). However, it seems that a different mechanism may underpin the cryoprotection that they confer to the cold-labile yeast frataxin homolog-1 (Yfh1). Circular dichroism spectroscopy (CD) was used to assess the degree of helicity of Yfh1 at 1 °C, both alone and in the presence of several dehydrin constructs. Three constructs were compared to the wild type: YSK_2_-K→R (lysine residues substituted with arginine), YSK_2_-Neutral (locally neutralized charge), and YSK_2_-SpaceK (evenly distributed positive charge). The results show that sequence rearrangements and minor substitutions have little impact on the ability of the dehydrin to preserve LDH activity. However, when the positive charge of the dehydrin is locally neutralized or evenly distributed, the dehydrin becomes less efficient at promoting structure in Yfh1 at low temperatures. This suggests that a stabilizing, charge-based interaction occurs between dehydrins and Yfh1. Dehydrins are intrinsically disordered proteins, expressed by certain organisms to improve desiccation tolerance. These proteins are thought to serve many cellular roles, such as the stabilization of membranes, DNA, and proteins. However, the molecular mechanisms underlying the function of dehydrins are not well understood. Here, we examine the importance of positive charges in dehydrin sequences by making substitutions and comparing their effects in the cryoprotection of two different proteins.

## 1. Introduction

The plant kingdom has developed robust defense mechanisms to protect against many abiotic stresses, such as dehydration, which can be caused by drought and low temperatures. During water loss, plants face multiple problems, such as compromised membranes, increased concentration of reactive oxygen species, and a weakened hydrophobic effect [1]. One component of the dehydration defense mechanism is a group of intrinsically disordered proteins known as the late embryogenesis abundant (LEA) proteins, which includes the dehydrins studied here. LEA proteins were first identified in *Gossypium hirsutum* during late embryogenesis, a growth stage marked by naturally low water content [2]. They have since been found throughout the plant kingdom, not only during late embryogenesis but in vegetative tissue as well [3]. The role of LEA proteins has been studied in a number of transgenic plants through overexpression and knockout experiments [4,5,6,7,8,9,10]. Under cold and/or drought conditions, these plants display enhanced growth and survival, and reduced electrolyte leakage and lipid peroxidation compared to their non-transgenic counterparts. While the mechanism by which LEA proteins accomplish this in vivo is uncertain, in vitro evidence has shown that LEA proteins are able to scavenge ions and stabilize biomacromolecules [3,11,12,13].

Several groups of LEA proteins exist, defined by the presence of conserved sequence motifs. Among the most studied LEA proteins are those in group 2, also known as dehydrins (dehydration-induced proteins). Dehydrins are defined by the presence of a lysine-rich sequence motif known as the K-segment ([XKXGXX(D/E)KIK(D/E)KXPG], where X is any amino acid) [14]. In addition to this segment, dehydrins may have Y- and/or S-segments named after a central aromatic residue, such as tyrosine, and a string of 5–7 serine residues, respectively. As motifs, their sequence can be described as [D(D/E)(Y/H/F)GNPX] and [LHR(S/T)GS_4–6_(S/D/E)(D/E)_3_], respectively. More recently, an F-segment has been identified [15,16]. Like the Y-segment, the F-segment possesses aromatic residues ([EXXDRGXFDFX(G/K)]). These conserved segments are separated by poorly conserved ϕ-segments, which are rich in glycine and other polar amino acids, but are low in non-polar amino acids and generally lack cysteine. Dehydrins can be sorted into seven sequence architecture groups: Y_n_SK_n_, F_n_SK_n_, Y_n_K_n_, SK_n_, F_n_K_n_, K_n_, and K_n_S. These groups are representative of which segments are present and how many of each are present is denoted by the ‘n’.

Dehydrins are overwhelmingly polar and lack the hydrophobic core that necessitates folding in ordered proteins. Instead, they are intrinsically disordered proteins (IDPs), specifically of the random coil type [12,17]. Random coil IDPs have no stable secondary structure and their tertiary structure is said to resemble spaghetti in boiling water [18]. The high glycine content of dehydrins give them considerable flexibility to randomly sample different conformations, limited mainly by backbone constraints and electrostatic repulsion. Their flexibility allows them to form very specific interactions with a multitude of binding partners. However, this comes at the cost of a significant loss of entropy, which makes these interactions highly reversible [19]. This, coupled with IDPs not having folded structure to be lost under denaturing conditions, makes dehydrins well suited as chaperones. One proposed role is as a macromolecule stabilizer under cold stress.

The cold denaturation of proteins is thought to occur when the surrounding water loses enough entropy that the entropic gain of protein unfolding outweighs the entropic loss of water organizing around exposed hydrophobic residues [20]. Lactate dehydrogenase (LDH) is commonly used to test the ability of a dehydrin to prevent cold-induced denaturation [17,21,22,23,24,25,26,27,28,29,30,31,32,33,34,35,36,37]. Typically, cryoprotection is measured by calculating the PD_50_ value (the concentration of additive required to recover 50% of the untreated LDH activity), where lower values represent more efficient protein cryoprotection.

Several studies have attempted to dissect the role of dehydrin segments in cryoprotection. Some studies have implicated the K-segment because a decrease in cryoprotective efficiency was observed when a K-segment was deleted [30,32]. Others have studied the cryoprotective efficiency of K-segment peptides and found it to be reduced when the hydrophobic residues were substituted with hydrophilic residues [35,37]. However, it appears likely that the greatest factor in cryoprotective efficiency, at least where LDH is concerned, is hydrodynamic radius [33]. When the ϕ-segment of a K_2_ dehydrin was removed, the dehydrin became a less effective cryoprotectant of LDH, despite retaining the same number of K-segments [31]. A follow-up study found that the cryoprotection of LDH by dehydrins could be largely explained as a relationship between the hydrodynamic radius and PD_50_ [33]. Additionally, polyethylene glycol (PEG), the disordered polar polymer, displays a similar relationship between hydrodynamic radius and cryoprotective efficiency [33].

One drawback to using LDH in cryoprotection assays is that it aggregates under cold stress, making it difficult to study the unfolding and protection processes. This prompted us to seek out an additional cold-labile protein for dehydrin cryoprotection assays that could be performed; the candidate that was used was yeast frataxin homolog 1 (Yfh1), a model protein for studying the cold denaturation of proteins [38]. Under physiological conditions, Yfh1 can begin to denature at 7 °C, such that denaturation studies can be carried out above freezing temperatures and without the need for destabilizing agents [38]. With respect to Yfh1, we found that the cryoprotective effects of dehydrins interestingly did not align with those observed when using LDH [36]. When the sequence of the dehydrins being studied was randomly scrambled, the scrambled variants were not necessarily as efficient cryoprotectants of Yfh1 as their non-scrambled counterparts, and two dehydrins were similarly efficient in their cryoprotection of Yfh1, despite having a three-fold difference in their hydrodynamic radii.

Here, we examine the importance of YSK_2_ sequence order in a dehydrin in its contribution to the cryoprotection, with an emphasis on the distribution of positive charges. Variants of the YSK_2_ dehydrin from *Vitis riparia*, a model dehydrin in our research, were constructed with substitutions and strategically rearranged sequence, and cryoprotection was studied using LDH in a freeze/thaw assay and Yfh1 in a low temperature denaturation assay.

## 2. Materials and Methods

### 2.1. Dehydrin Construct Design and Cloning

Protein sequences were submitted to Invitrogen GeneArt to be optimized for expression in *E. coli*. The genes were subsequently subcloned into pET-22b(+) (Novagen) and the success of the digestion and ligation were confirmed by DNA sequencing. The YSK_2_ gene, courtesy of Dr. Annette Nassuth (University of Guelph, Guelph, ON, Canada), had been previously cloned into pET22b-YSK_2_ as described in Livernois et al. [39]. Note that the sequence used here contains a H63N substitution relative to the UniProt sequence Q9M605. The pETYF-1 plasmid was a generous gift from Dr. Grazia Isaya (University of Minnesota Rochester, Rochester, MN). This plasmid encodes a truncated version of the mature Yfh1 (residues 54–174) and had previously been subcloned into pET24a(+) (Novagen) as described by Palmer et al. [36].

### 2.2. Dehydrin Expression and Purification

The YSK_2_ and YSK_2_ constructs were expressed and purified using a slight modification of Livernois et al. [39]. Briefly, the cells were lysed by boiling for 20 min with agitation every 5 min and were subsequently cooled on ice. Sodium acetate (pH 5.0) was added to a final concentration of 20 mM and the supernatant was filtered before continuing with purification. The filtered supernatant was loaded onto a 5 mL HiTrap SP HP column (Cytiva, Vancouver, BC, Canada) equilibrated with 20 mM sodium acetate, pH 5.0 (buffer A). Elution occurred over a gradient of 0–1 M NaCl (Buffer B: 20 mM sodium acetate, pH 5.0, 1 M NaCl). Fractions containing the protein of interest were desalted into Milli-Q water using a HiPrep 26/10 column (Cytiva). After concentrating the eluted protein fraction, reversed-phase HPLC was performed. The protein sample was purified using a BioBasic C4 column (Thermo Scientific, Ottawa, ON, Canada), which was equilibrated with Buffer A (0.1% trifluoroacetic acid (TFA) (*w*/*v*)). The protein was eluted using a 1–100% Buffer B (0.1% TFA (*w*/*v*) in acetonitrile) gradient. Sample purity was determined using 12% SDS-PAGE and the fractions containing the protein of interest were pooled, flash frozen and lyophilized. The proteins were stored at −20 °C until use.

### 2.3. Yfh1 Expression and Purification

Yfh1 was purified as described by Palmer et al. [36] Briefly, plasmid containing cells were grown at 37 °C to an OD_600_ of 0.6–0.8 prior to the addition of 0.4 mM IPTG. The cells were induced overnight at 16 °C with shaking. After centrifugation at 6000× *g* for 15 min, the media were decanted and the cell pellets were resuspended in 20 mM Tris, pH 8.0, 50 mM NaCl with an EDTA-free protease inhibitor tablet (Roche, Mississauga, ON, Canada) and 0.5 mM EDTA. Cells were lysed by sonication. The supernatant was collected by centrifugation at 20,000× *g* for 20 min at 4 °C, and was subsequently filtered through a 0.22 μM PVDF membrane. The filtered supernatant was run through a HiTrap Q FF anion exchange column (Cytiva) equilibrated with Buffer A (20 mM Tris HCl, pH 8, 50 mM NaCl) and eluted over a linear NaCl gradient of 50–600 mM (Buffer B: 20 mM Tris HCl, pH 8, 600 mM NaCl). Fractions containing Yfh1 were identified by 12% SDS-PAGE, then pooled, concentrated to ~3 mL, and passed through a Superdex 75 10/300 GL column (Cytiva) equilibrated with 10 mM Tris, pH 7.4, 100 mM NaCl buffer. Fractions containing the protein of interest were pooled and desalted into 10 mM Tris, pH 7.4, at room temperature using a HiPrep 26/10 desalting column (Cytiva). The Yfh1 was then concentrated to 3 mL using an Amicon ultra 15 mL centrifugal filter (10,000 Da molecular weight cut-off; Millipore, Etobicoke, ON, Canada), aliquoted, flash frozen, and stored at −20 °C until use.

### 2.4. LDH Cryoprotection

The LDH cryoprotection experiment was performed as previously described by Hughes and Graether [31] with small modifications. Briefly, 100 μg/mL of LDH from rabbit muscle (Sigma-Aldrich, Oakville, ON, Canada) was suspended in 10 mM Na_2_HPO_4_, pH 7.4, to a final volume of 500 μL, dialyzed overnight at 4 °C against 2 L of 10 mM Na_2_HPO_4_, pH 7.4, and lastly diluted twofold. The wild-type YSK_2_ and the dehydrin constructs (YSK_2_-SpaceK, YSK_2_-Neutral, and YSK_2_-K→R) had their concentrations determined using the BCA assay (Pierce, Ottawa, ON, Canada), and were then adjusted to a range of concentrations (5, 20, 50, 80, 120, 560, and 1000 μg/mL). LDH (to a final concentration of 5 μg/mL) and dehydrin were combined, 8 μL each, and all experiments were performed in triplicate. Each sample was then flash frozen for 30 s and thawed at 16 °C for 5 min. The cycle was repeated a total of five times for each sample. A volume of 724 μL of reaction buffer (680.6 μL of 30 mM Na_2_HPO_4_, pH 7.4, 21.7 μL of 2 mM NADH, and 21.7 μL of 10 mM pyruvic acid) was then added to each sample immediately before measuring A_340_ over the course of 2.5 min, with readings being taken each second using a Cary 100 UV-Vis spectrophotometer. The percent LDH activity was found by dividing the initial reaction velocity of the frozen samples by that of the positive control that had not been frozen. The data were plotted as %LDH activity vs. log(dehydrin concentration). The result was a sigmoidal curve fitted using %LDH activity = a/(1 + e^(−(x−xo)/b^)), where x is the dehydrin concentration and a, b, and x_o_ were fitted variables.

### 2.5. Circular Dichroism

Prior to analysis, dehydrin samples were dissolved in 10 mM Tris, pH 7.4, and dialyzed overnight at 4 °C against 10 mM Tris, pH 7.4, to remove any contaminating salts. The dialyzed samples were filtered through an Ultrafree 0.22 μM PVDF membrane (Millipore). The Pierce BCA Protein Assay was used in conjunction with a BSA standard curve to determine dehydrin concentration. Yfh1 concentration was detected using A_280_ with an extinction coefficient of 15,470 M^−1^ cm^−1^, as predicted by the ExPASy server.

Circular dichroism (CD) was performed using a JASCO-815 spectropolarimeter (Jasco Inc., Easton, MD) and quartz cuvettes with 1 mm pathlengths. Spectra were collected from 190–250 nm using a bandwidth of 1 nm and a scan speed of 50 nm/min. Six accumulations were collected for each sample. Spectra of 10 μM Yfh1 were collected at 25 °C and 1 °C. Spectra of 10 μM Yfh1 in the presence and absence of YSK_2_, YSK_2_-SpaceK, YSK_2_-Neutral, and YSK_2_-K→R collected over a range of dehydrin concentrations (5, 7.5, 10, 15, 20, and 25 μM). The contribution of the dehydrins to the CD signal was subtracted, and the percent α-helicity of Yfh1 was then estimated using %α-helicity = 100 × (3000 − [θ]_222_)/39000, where [θ]_222_ is the molar ellipticity of Yfh1 at 222 nm in deg × cm^2^/dmol [40].

### 2.6. NMR Experiments

^15^N-HSQC-NMR spectra of ^15^N-YSK_2_ were collected in the presence and absence of Yfh1. The expression and purification of ^15^N-YSK_2_ was performed as described above with one exception. Upon reaching an OD_600_ of 0.6–0.8, the cells were washed and resuspended in M9-minimal media [39] containing ^15^N-NH_4_Cl, and then induced with IPTG. Samples were suspended in NMR buffer (50 mM Na_2_HPO_4_, 10 mM NaCl, pH 6.0, 0.01% (*w*/*v*) sodium azide, 0.1 mM DSS and 5% D_2_O (*v*/*v*)). One sample contained 0.5 mM ^15^N-labelled YSK_2_ alone, and the other contained 0.5 mM of ^15^N-labelled YSK_2_ and 0.5 mM of unlabeled Yfh1. Both experiments were performed at 300 K. Spectra were collected on a Bruker Avance DRX600 spectrometer with a cryogenic probe. Chemical shift assignments of YSK_2_ were obtained from Findlater and Graether [41]. The data were processed using NMRPipe v11.1 software [42] and analyzed with CCPNMR v2.4 [43].

## 3. Results

### 3.1. Construct Design and Structure

Three constructs were designed based on the wild-type sequence of YSK_2_ (Figure 1). The YSK_2_-K→R construct had all lysine residues substituted with arginine to examine the importance of positive charge versus the need for a lysine residue (Figure 1A). The YSK_2_-SpaceK and YSK_2_-Neutral constructs have the same amino acid composition as YSK_2_; however, the residues have been rearranged (Figure 1B). In the YSK_2_-SpaceK, the lysine residues are spaced evenly throughout the sequence, separated by ten residues, such that the positive charge is no longer clustered in the K-segments. In YSK_2_-Neutral, the aspartate and glutamate residues have been moved to be next to the lysine residues in the K-segments to locally neutralize the charge.

With the introduction of a large number of changes in an IDP sequence, there is a concern that the level of disorder may inadvertently change, which could affect the cryoprotection assay results [36]. To ensure that the constructs were similarly disordered as the wild-type YSK_2_, the CD spectra were collected and compared (Figure 2). All spectra displayed attributes characteristic of disordered proteins, including very little ellipticity above 210 nm and a minimum near 200 nm [44]. The spectra are very similar in shape between the four proteins, with all minima centered at 198 nm. The only exception was the molar ellipticity values, which were −14,000 deg·cm^2^/dmol for YSK_2_, YSK_2_-SpaceK, and YSK_2_-K→R, with YSK_2_-Neutral having a minimum of −16,000 deg·cm^2^/dmol, suggesting that this construct may be slightly more disordered. Based on these results, it is assumed that the degree of disorder of the constructs should not be a conflating variable.

### 3.2. LDH Cryoprotection

We first examined the effects the constructs may have in the LDH cryoprotection assay. The treatment of LDH with multiple freeze/thaw cycles results in very little enzymatic activity while the addition of proteins results in the conservation of different levels of activity. The results are shown in Figure 3 over a range of protein concentrations as a percent conservation of the untreated LDH activity. All four dehydrins had similarly shaped curves, with different enhancements of LDH activity relative to the LDH control at high dehydrin concentrations. The wild-type YSK_2_ showed the lowest value, at about 130%. YSK_2_-SpaceK plateaued at about 140%, while YSK_2_-K→R and YSK_2_-Neutral did so at about 150%. BSA, a known cryoprotectant [45], only increased LDH activity to 120% of the control level. Therefore, the degree to which these constructs retained the original LDH activity was slightly better than the wild-type YSK_2_ and BSA. The fits to the data (described in Section 2) were then used to extract the PD_50_ values of all of the proteins. Each of the dehydrin constructs had a similar PD_50_ of 2 to 3 μM, while BSA had a PD_50_ of 0.8 μM, showing that the efficiency of protection by the constructs was essentially unaffected by the mutations.

### 3.3. Yfh1 Secondary Structure

Yfh1 was selected for this study because it begins to denature at temperatures starting at 7 °C at low salt concentrations [38] and does not aggregate, therefore making it amenable to spectrophotoscopic techniques. We compared the α-helical structure of Yfh1 at 25 °C and 1 °C using CD spectroscopy. When suspended in 10 mM Tris (pH 7.4), the CD spectrum of Yfh1 acquires characteristics associated with disorder at 1 °C that were not present at 25 °C (Figure 4). At 25 °C, Yfh1 is predominantly α-helical, with minima at 208 and 222 nm and a maximum around 190 nm [46], reflecting an α-helical content of 35%. At 1 °C, however, the ellipticity above 210 nm decreases significantly, and the minimum shifts to 201 nm. At 1 °C the ellipticity at 222 nm corresponds to an estimated helical content of 24%. The loss of structure at 1 °C was reversed when the temperature was raised.

### 3.4. Yfh1 Secondary Structure in the Presence of Dehydrins

We next examined the ability of YSK_2_ and the constructs to restore Yfh1 secondary structure at several different concentrations of the constructs. The wild-type YSK_2_ and the YSK_2_-K→R were the most effective at enhancing α-helicity in Yfh1 at 1 °C (Figure 5A,B). Both proteins increased helicity at each concentration tested; the minima at 208 and 222 nm became increasingly negative, and the maximum at 190 nm became increasingly positive. In contrast, YSK_2_-SpaceK and YSK_2_-Neutral were able to increase the Yfh1 helical content, though to a lesser extent (Figure 5C,D). Spectral data for the 25 μM sample were excluded due to excessive noise at <205 nm. The retention of secondary structure for YSK_2_-SpaceK (Figure 5D) required 25 μM of protein to maintain helicity to the same level as Yfh1 at 25 °C, while the YSK_2_-Neutral construct was not able to maintain Yfh1 structure at 1 °C to the same level as Yfh1 alone at 25 °C.

We next calculated the percent helicity of Yfh1 in the presence of these proteins (Figure 6). In their absence, Yfh1 was 24% α-helical at 1 °C, and 35% at 25 °C. The addition of dehydrins caused the helicity to increase with increasing concentration, though the effect plateaued at different levels. As seen in the figure, the most effective proteins were YSK_2_ and YSK2-K→R. With the addition of 7.5 μM YSK_2_, the spectrum of Yfh1 at 1 °C was nearly indistinguishable from that of Yfh1 at 25 °C, whereas at 7.5 μM the YSK_2_-K→R construct already increased the helicity beyond that of Yfh1 at 25 °C by 2.5%. At each concentration of the dehydrins, the total helical content of Yfh1 continued to increase significantly. The α-helicity of Yfh1 in the presence of 25 μM of YSK_2_ starts to plateau at ~50% helical content. The YSK_2_-K→R construct reached an even higher helical content of 55%, but instead of plateauing began to decrease the helical content to 49% at 25 μM of dehydrin.

The other two constructs, YSK_2_-SpaceK and YSK_2_-Neutral, were not as effective at inducing helicity. While YSK_2_-SpaceK increased the helicity of Yfh1 at 1 °C at each concentration, only the 25 μM Yfh1 protein at 1 °C was restored to the same helicity as the protein at 25 °C, and even then, the effect appeared to plateau. The effect of YSK_2_-Neutral plateaued at 28%, and therefore had the weakest protective effect of all of the constructs.

We performed experiments to examine whether there are cryoprotective mechanisms protecting LDH and Yfh1 in common from cold damage caused by a temperature of 1 °C. In the first experiment, we examined whether polyethylene glycol 10,000 (PEG 10,000) is able to effectively protect Yfh1 as it does LDH [36]. The results here show that PEG 10,000, which has a similar hydrodynamic radius (R_h_) to YSK_2_, was completely ineffective at promoting helicity in Yfh1 at 1 °C (Figure 7); there was essentially no difference in the spectra in the presence or absence of this polymer as measured by CD. Next, we examined whether Yfh1 is protected by dehydrins without a binding interaction, as is the case of LDH cryoprotection [31]. We collected ^15^N-HSQC spectra of ^15^N-labelled YSK_2_ alone and in the presence of an equimolar amount of unlabeled Yfh1 (Figure 8). As the spectra show, many resonances were unaffected by the presence of Yfh1, but several small changes in chemical shift were observed.

## 4. Discussion

### 4.1. Dehydrin Construct Design

Prior experiments with LDH showed [31,33] that the cryoprotective effects for it can be largely explained by a molecular shield effect [47,48]. However, using another protein that is sensitive to cold denaturation, Yfh1, our previous study showed that the size of the dehydrin is not a factor in cryoprotection, suggesting that the protective shield effect is not a universal one [36]. To examine the possibility that dehydrins confer cryoprotection to Yfh1 in a way that is distinct from LDH, three constructs were designed to probe the significance of positive charge, since lysine residues are highly conserved in dehydrins (Figure 1). YSK_2_-SpaceK and YSK_2_-Neutral have an amino acid composition identical to that of YSK_2_, but specific residues have been moved to either distribute the positively charged lysines evenly throughout the sequence (YSK_2_-SpaceK) or to offset the charge of the lysine residues by moving the aspartates and glutamates adjacent to the lysines (YSK_2_-Neutral). In both cases, the expectation was that the changes may interrupt a charge-based interaction between the dehydrins and Yfh1, which would in turn result in poorer cryoprotection. The other construct (YSK_2_-K→R) had all lysine residues substituted with arginine. The creation of the YSK_2_-K→R construct sought to address whether lysine itself is important in this cryoprotective effect, or if it is merely the distribution of positive charge within the K-segments. These lysine residues are well conserved and are not substituted for arginine in wild-type dehydrins, implying that they serve some purpose that arginine cannot fulfill [14].

### 4.2. Yfh1 Cryoprotection Is Affected by Charge Distribution

As has been previously reported [38], the lowering of the temperature results in Yfh1 losing secondary structure, mainly observed as a loss in α-helicity, and we observed similar results here (Figure 4). At 1 °C, the Yfh1 CD spectrum took on characteristics associated with disorder, such as a large decrease in ellipticity above 210 nm and a shift in the minima from 207 to 201 nm. We exploited this property to see whether a dehydrin can act as a cryoprotectant by protecting the structure of Yfh1 by maintaining its α-helicity.

The temperature instability of Yfh1 has been attributed to a cluster of negatively charged residues (Glu89, Glu103, Glu112) that sit on adjacent β-strands and cause steric repulsion, allowing water access to the hydrophobic core and causing the onset of cold denaturation at temperatures above 0 °C [49]. Thus, a cluster of positively charged residues, similar to what is seen in the K-segments, could potentially allow for an interaction that alleviates some of that charge repulsion, thereby stabilizing Yfh1. Conversely, the absence of concentrated positive charge, due to it being more distributed or locally neutralized, could disrupt that interaction. We hypothesized that YSK_2_-SpaceK and YSK_2_-Neutral would be less effective cryoprotectants than YSK_2_ and YSK_2_-K→R based on the assumption that the distribution of positive charge is a significant part of the cryoprotective mechanism with respect to Yfh1 but not whether it is a lysine or arginine. This was demonstrated by the results (Figure 5 and Figure 6). With YSK_2_-Neutral and YSK_2_-SpaceK the helicity of Yfh1 maximally increased to ~30 and 35%, respectively, while YSK_2_ and YSK_2_-K→R maximally increased the helicity to ~50% or more.

Surprisingly, the YSK_2_-K→R protein was more effective at protection than the wild-type dehydrin (Figure 6), suggesting that positive charge is more significant than the lysines themselves, and that the potential involvement of the K-segment is coincidental rather than a product of evolutionary design. This is reasonable because Yfh1 is a yeast protein that would not come in contact with plant dehydrins in the natural world. However, dehydrins are often cationic, and within eukaryotic cells, the majority of proteins are anionic [50], suggesting that the charge of dehydrins has evolved to protect many proteins.

With YSK_2_-K→R appearing to be more effective at promoting helicity in Yfh1 than YSK_2_, the question arises as to why lysine residues are conserved in K-segments instead of arginine. However, by 25 μM YSK_2_-K→R, the helicity of Yfh1 began to decrease compared to 20 μM. One possible reason for this is that arginine has more potential to hydrogen bond compared to lysine, which has been cited as useful for protein stabilization [51]. On the other hand, the hydrogen bonding potential coupled with the planar structure of arginine makes it capable of self-association in a way that lysine, which has a tetrahedral structure, cannot [52]. Arginine is able to form hydrogen bonds with the protein backbone, and the guanidinium group is able to form ionic interactions with itself and participate in π–π stacking; by contrast, lysine residues repel each other and do not perform π–π stacking [52,53,54]. It could be argued that the arginine residues allow YSK_2_-K→R to be a marginally more efficient cryoprotectant towards Yfh1 at low concentrations, but at high concentrations the K→R construct could self-associate and the cryoprotection decreases. Another possibility is that the strong conservation of lysine in the K-segments could be significant for another function, such as membrane binding [55], and that the positive charges could incidentally also be advantageous for forming stabilizing interactions with various acidic proteins during cold stress.

YSK_2_-SpaceK and YSK_2_-Neutral were significantly less efficient at increasing the helicity of Yfh1 at cold temperatures (Figure 6). For YSK_2_-SpaceK, this is likely due to the different charge distributions compared to YSK_2_. If the dehydrins stabilize Yfh1 by relieving electrostatic repulsion, then having the positive charge spread throughout the dehydrin may reduce the effectiveness of this interaction since fewer lysines would be able to participate in a binding interaction with a local cluster of negative charges in the protected protein. For YSK_2_-Neutral, the negatively charged residues could offset the influence of the lysines by neutralizing the local charge, possibly preventing any interaction with the cluster of negative charges on Yfh1. This could explain why it had the least amount of protection.

### 4.3. Yfh1 Is Not Protected by a Molecular Shield Effect

In a previous study we showed that in the LDH cryoprotection assay PEG molecules of a similar R_h_ to the dehydrins had similarly efficient levels of recovered activity [33]. PEG may stabilize proteins by coordinating with positive residues and binding to hydrophobic clefts [56,57]. The encapsulating layer of PEG molecules can prevent the proteins from coming into contact with others and causing aggregation. None of these stabilizing interactions would alleviate negative electrostatic repulsion in Yfh1 under the conditions used for this experiment because Yfh1 was not aggregating. It is not surprising that PEG 10,000 was unable to protect Yfh1 from cold denaturation (Figure 7), and therefore the molecular shield effect is not at play.

We provide here a second piece of evidence that the molecular shield effect is not important. Previous NMR studies have shown that there is no interaction between K_2_ and LDH [31]. Even using a threefold molar excess of LDH, no change in chemical shifts was observed in the ^15^N-K_2_ spectrum. The ^15^N-HSQC-NMR spectrum of labelled YSK_2_ in the presence of Yfh1 (Figure 8) revealed that in contrast to LDH, there is a weak interaction as demonstrated by several small shifts in the peaks of YSK_2_. Nearly all of these shifts are found in the K-segments, suggesting that the interaction may be located in this conserved motif. In fact, the only significant shifts that were not part of the K-segment were residues located in the ϕ-segment between the two K-segments.

## 5. Conclusions

The results presented here show that dehydrins protect Yfh1 in a different manner than they do LDH, and that dehydrins likely contribute to more than one mechanism of cryoprotection. This is demonstrated by all constructs having a similar PD_50_ when used to preserve LDH activity in the freeze/thaw assay, emphasizing the importance of R_h_ over sequence order. In the Yfh1 cryoprotection, the YSK_2_ constructs with more dispersed (YSK_2_-SpaceK), or locally neutralized positive charge (YSK_2_-Neutral) were less effective at promoting helicity in Yfh1 at cold temperatures than the proteins with clustered positive charges (YSK_2_ and YSK_2_-K→R). This suggests an interaction between the dehydrins and Yfh1 that may help to alleviate the destabilizing negative charge cluster. The ^15^N-HSQC NMR spectrum of YSK_2_ in the presence of Yfh1 supports the idea that a weak interaction may be taking place. The data suggest that future studies must aim to better characterize the binding interaction.

## Figures and Tables

**Figure 1 biomolecules-12-01510-f001:**
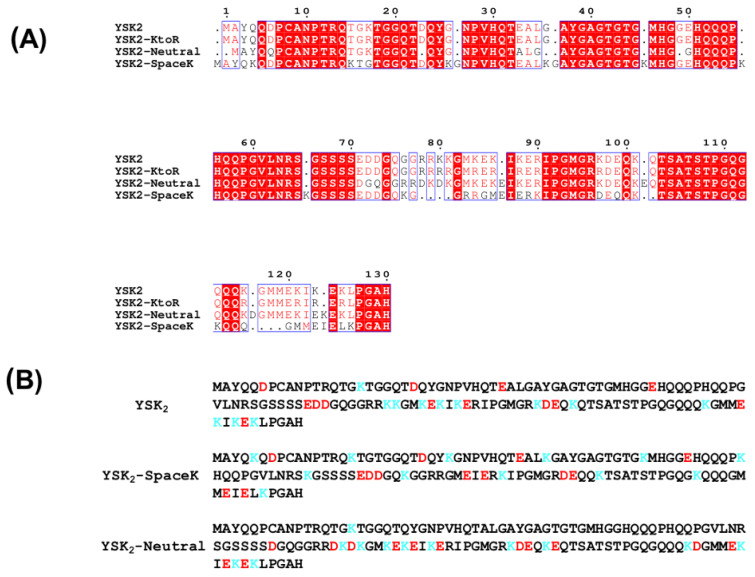
Sequences of wild-type YSK_2_ and the three constructs. (**A**) Multiple sequence alignment of YSK_2_, YSK_2_K→R, YSK_2_-SpaceK, and YSK_2_-Neutral. The alignment was performed using ClustalW and the image was created using ESPript. Conserved regions are shown bounded by a blue box. White letters on a red background show residues conserved among all sequences, while red letters on a white background show residues that are similar. (**B**) Distribution of lysine and negatively charged residues in the YSK_2_-SpaceK and YSK_2_-Neutral constructs. All negatively charged residues are shown in red and all lysine residues are shown in blue.

**Figure 2 biomolecules-12-01510-f002:**
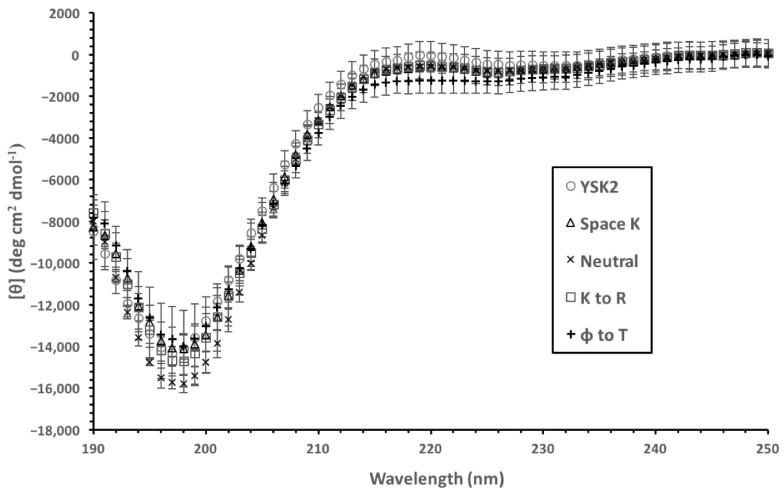
Comparison of the disorder of the wild-type YSK_2_ and the constructs. CD spectra of 10 μM protein in 10 mM Tris, pH 7.4, were collected at 25 °C. Symbols are shown as an inset. Each spectrum is an average of three replicates with six accumulations each. Error bars represent the standard deviation.

**Figure 3 biomolecules-12-01510-f003:**
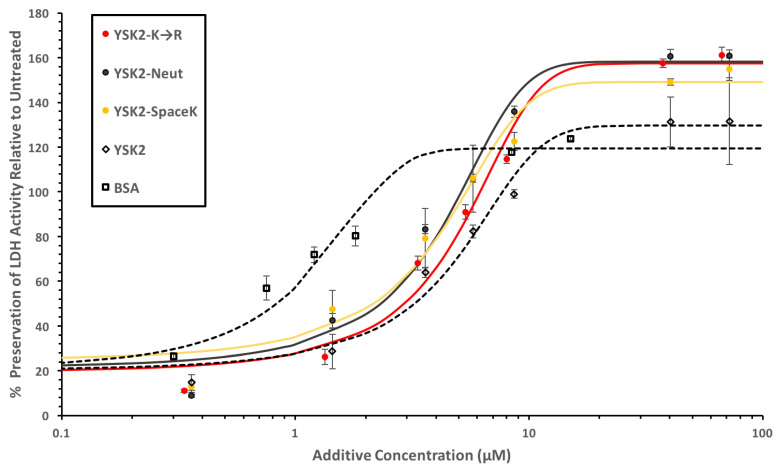
Cryoprotection of LDH by the YSK_2_ constructs. The LDH assay was performed as described by [31]. The legend is shown as a figure inset. Lines were fitted using the equation described in Section 2, where the percent LDH preservation is relative to untreated LDH in the absence of additives. Red line, YSK_2_-K→R; solid black line, YSK_2_-Neutral; YSK_2_-SpaceK, yellow line; YSK_2_, dashed line with diamond symbols; BSA, dashed line with square symbols.

**Figure 4 biomolecules-12-01510-f004:**
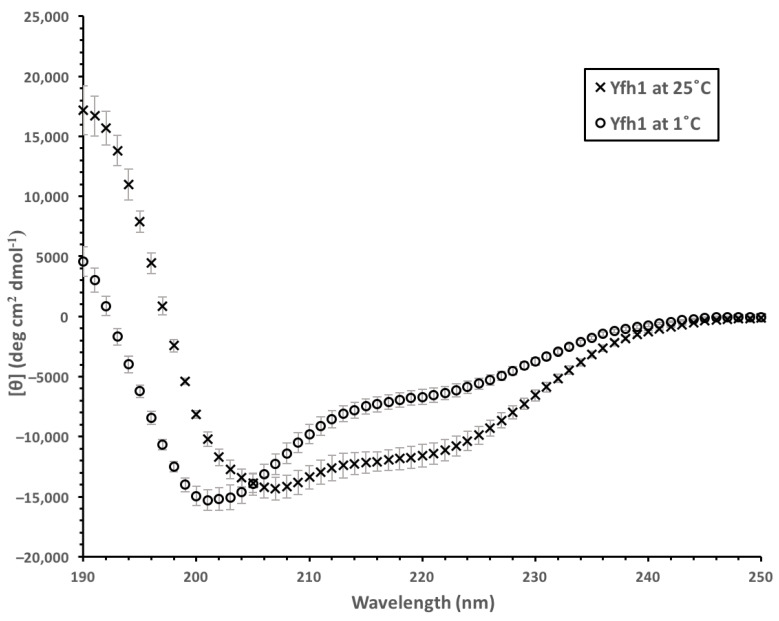
Spectra of Yfh1 at 25 and 1 °C. CD spectra of 10 μM Yfh1 at 25 °C and 1 °C in 10 mM Tris, pH 7.4. The spectra shown here are an average of eight replicates with error bars representing standard deviation.

**Figure 5 biomolecules-12-01510-f005:**
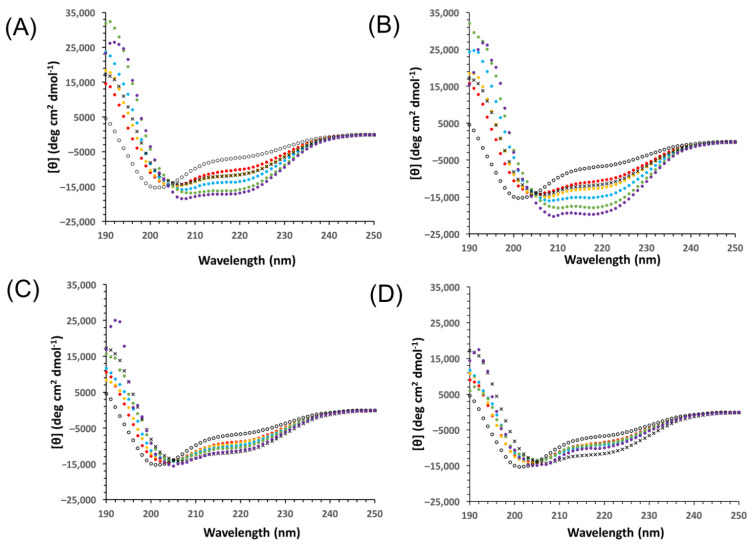
Yfh1 spectra in the presence of YSK_2_ constructs. CD spectra of 10 μM Yfh1 at 1 °C in the presence of 5 (red), 7.5 (yellow), 10 (blue), 15 (green), and 20 μM (purple). (**A**) Yfh1 spectra in the presence of YSK_2_, (**B**) Yfh1 spectra in the presence of YSK_2_-K→R, (**C**) Yfh1 spectra in the presence of YSK_2_-SpaceK and (**D**) Yfh1 spectra in the presence of YSK_2_-Neutral. The panels also contain the spectra of 10 μM Yfh1 alone at 1 °C (black circles) and at 25 °C (black × symbols). Each spectrum is an average of three replicates. All samples were in 10 mM Tris, pH 7.4, buffer.

**Figure 6 biomolecules-12-01510-f006:**
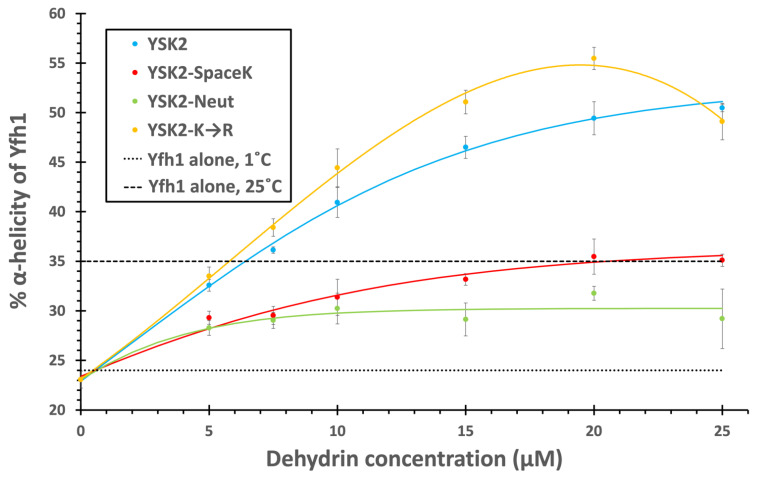
The Relationship between Yfh1 helicity and construct concentration. The CD spectrum of 10 μM Yfh1 was determined at 1 °C in 10 mM Tris, pH 7.4, in the presence and absence of the various dehydrin constructs. The equation described in Section 2 was used to find the percent α-helicity of Yfh1. Three replicates were used to create this image and the standard deviation is represented by the error bars. The dashed line indicates the helicity of Yfh1 alone at 25 °C, while the dotted line indicates the helicity of Yfh1 alone at 1 °C. YSK_2_, light blue line; YSK_2_-SpaceK, red line; YSK_2_-Neutral, green line; YSK_2_-K→R, yellow line.

**Figure 7 biomolecules-12-01510-f007:**
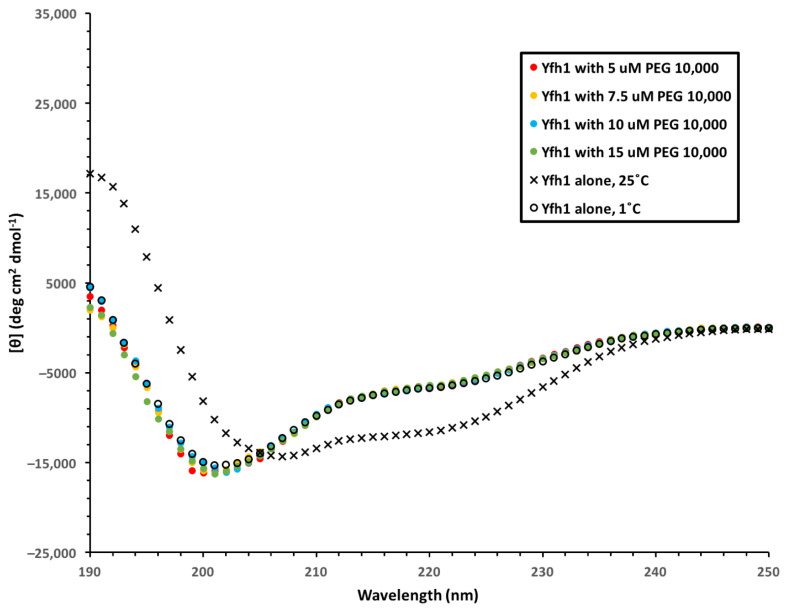
The effect of PEG 10,000 on Yfh1 Structure. CD spectra of 10 μM Yfh1 in the presence of various concentrations of PEG 10,000 at 1 °C in 10 mM Tris, pH 7.4.

**Figure 8 biomolecules-12-01510-f008:**
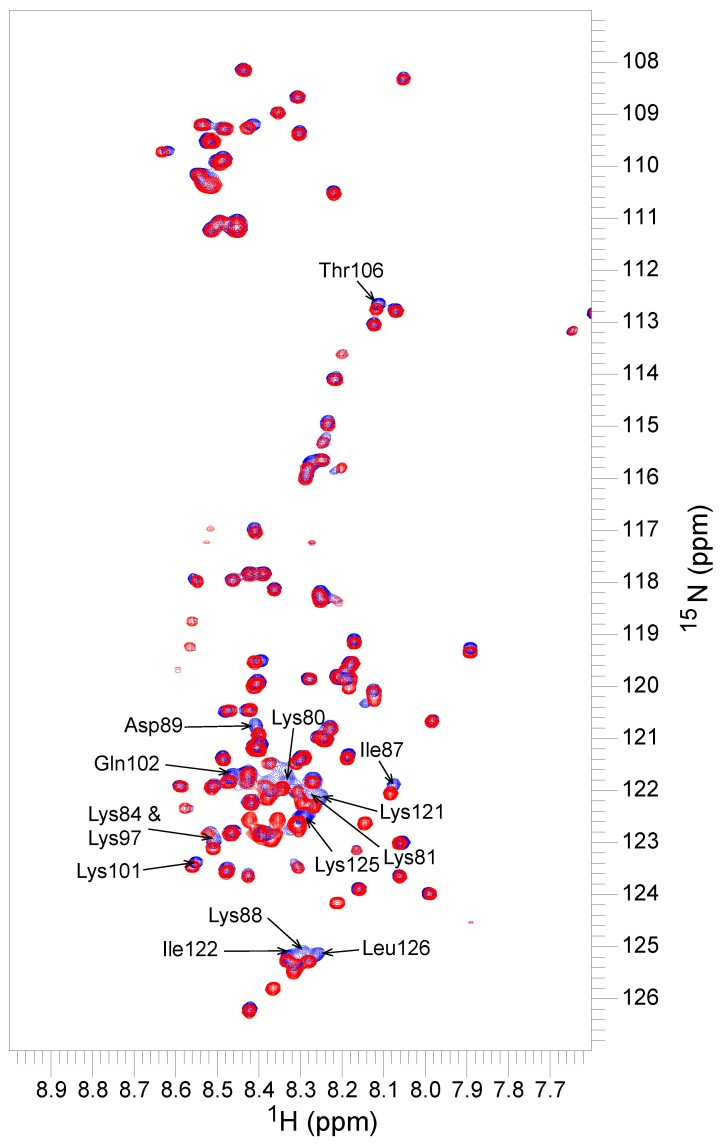
^15^N-HSQC spectra of YSK_2_ in the presence and absence of Yfh1. ^15^N-HSQC spectra overlap of 0.5 mM ^15^N-labelled YSK_2_ alone (red) and in the presence of 0.5 mM Yfh1 (blue). All samples are in pH 6.0 phosphate buffer. Data were collected at 300 K.

## Data Availability

Data available from the authors upon request.

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
