# Peer review of "The Effect of Positive Charge Distribution on the Cryoprotective Activity of Dehydrins"

_biomolecules, 2022, doi:10.3390/biom12101510_

Round 1
Reviewer 1 Report
The authors are to be commended for their research into the mechanism behind dehydrin protein protective function. Their use of the yeast protein Yfh1 is particularly inspired, circumventing, as it does, the issue of protein aggregation upon the commencement of denaturation.
I have only two issues of mediocre importance.
One concerning a lack of information about how the dehydrin was labeled with their N15 isotope (I could find no description). Another was their use of phosphate buffers for cooling/freezing experiments. The components of phosphate buffers (either Na or K as the cation) are notoriously insoluble at cold temperatures. Upon precipitation, pH is altered. This makes no difference for their conclusions, but the stress they hope to impose (cold) is now confounded with one that may be occurring due to pH changes. It is something they should consider.
Minor issues.
Their choice of descriptors for how the dehydrins were operating. Although it is clear that the authors are investigating dehydrin protein protective mechanisms, their choice of words implies they are looking at a repair mechanism. I have made suggestions throughout the attached PDF. To be clear, I have frequently put a line through text that I hope the authors will add to. I do not mean for them to remove their text in these instances but simply added text in the pdf is inconspicuous and frequently is overlooked. Hence my use of cross out. I trust that my meaning will be clear and that these instances will not be missed due to their conspicuous nature.

Author Response
We thank Reviewer 1 for the thoughtful corrections and comments, and hope we have addressed all of them as described below.
One concerning a lack of information about how the dehydrin was labeled with their N15 isotope (I could find no description).
We have added two sentences on lines 219-221 to describe how the protein was labelled.
Another was their use of phosphate buffers for cooling/freezing experiments. The components of phosphate buffers (either Na or K as the cation) are notoriously insoluble at cold temperatures. Upon precipitation, pH is altered. This makes no difference for their conclusions, but the stress they hope to impose (cold) is now confounded with one that may be occurring due to pH changes. It is something they should consider.
We agree that we should formally check whether using a different buffer changes the cryoprotection assay results. To our knowledge the use of phosphate buffer is standard in the field, and papers using other buffers (e.g. Tris, which also undergoes temperature dependent pH changes) have not shown significantly different results.
Their choice of descriptors for how the dehydrins were operating. Although it is clear that the authors are investigating dehydrin protein protective mechanisms, their choice of words implies they are looking at a repair mechanism. I have made suggestions throughout the attached PDF. To be clear, I have frequently put a line through text that I hope the authors will add to. I do not mean for them to remove their text in these instances but simply added text in the pdf is inconspicuous and frequently is overlooked. Hence my use of cross out. I trust that my meaning will be clear and that these instances will not be missed due to their conspicuous nature.
We completely agree with the reviewer that suggesting that there is a repair mechanism is incorrect, and have incorporated changes throughout the manuscript by using the word “preservation” to indicate that dehydrins are preserve activity rather than actively repairing it.
Reviewer 2 Report
The article describes a study aimed at investigating the mechanisms underlying the cryoprotective effect exerted by dehydrins on proteins that undergo cold denaturation, such as lactate dehydrogenase (LDH) and yeast frataxin homolog 1 (Yfh1). In particular, authors investigate the role of the positive charge in a YSK2-type isoform of dehydrin of Vitis riparia. Such isoform is composed by one Y- and S-fragment and two K-fragments that are conserved motifs rich in tyrosine, serine and lysine residues, respectively. To reach this goal the authors use YSK2 WT and different mutant constructs, called YSK2-K→R (lysine residues substituted with arginine), YSK2-Neutral (locally neutralized 15 charge), and YSK2-SpaceK (evenly distributed positive charge). The ability of dehydrin to prevent cold denaturation was evaluated by using biophysical techniques, in case of Yfh1, and enzymatic assays, for LDH, as suggest by literature. LDH, in fact, is a tetrameric enzyme that unfolds below 4 ° C and has been found to aggregate. Yeast frataxin is an important model to study cold denaturation, it unfolds below 7 ° C and since it does not aggregate it can be characterized by using biophysical methods. Previous articles have shown that the radius of the cryoprotective agent is important to restore the activity of LDH, here, authors principally studied the role of the positive charge on the cryoprotective effect on Frataxin. In particular, has been found that only YSK2 WT and KR are able to recover and also increase the helical content of Yfh1 in denaturant condition (1°C). In case of Yfh1, the radius does not seem to affect this capability; in fact, in presence of PEG, Yfh1 remains denatured at 1°C. 15N HSQC-NMR spectra of YSK2 WT alone and in presence of Yfh1 were acquired, thus finding that some residue, especially Lysine residues located in the K-fragments, are involved in an interaction.
The article is clear, well written. The literature was correctly cited, and the experiments correctly described. The experimental plan is clear and contains right controls. The results have been correctly interpreted and contextualized by the authors. The impact of the research is sectoral but of interest.
Here I report some comments and requests:
Line 162: Basing on the data provided, the LDH concentration used for the assay is unknown. Please indicate the initial volume or the final concentration used.
Line 279, the concentration discussed here (25uM) is not reported in graph 5D basing on the colors described in the legend. I suggest adding the corresponding spectrum in the graph or modifying the text.
- Although in the case of the LDH protein, the cryoprotective effect of YSK2 WT and constructs is comparable (PD50s are similar), most likely due to the equal radii of the various constructs, it is not clear why, in the case of the mutated constructs, the LDH activity is not only restored but also increased. Please comment this point.
- Please discuss the possible reasons of the increase in the alpha helix content in Yfh1 at 1°C induced by YSK2 ET and KR compared to the native form at 25 ° C.
- Furthermore, I checked the sequences of YSK2 WT and mutants and I noticed that the socalled WT shows mutation H63N (Q9M605, uniport). Indicate whether this is an error in the sequence reported or whether this mutation actually exists. In the latter case, please report it in Materials and Methods.
- If possible, also report the 15N-HSQC NMR spectra of YSK2 RK construct alone and in the presence of Yfh1.
Author Response
We thank Reviewer 2 for the thoughtful corrections and comments, and hope we have addressed all of them as described below.
Line 162: Basing on the data provided, the LDH concentration used for the assay is unknown. Please indicate the initial volume or the final concentration used.
We have added on Line 192 that the final concentration of LDH is 5 ug/mL.
Line 279, the concentration discussed here (25uM) is not reported in graph 5D basing on the colors described in the legend. I suggest adding the corresponding spectrum in the graph or modifying the text.
At high concentration of dehydrins the spectral signal is excessively noisy in the 190 - 200 nm range, obscuring the other data. We have indicated this in the text on line 330 with “Spectral data for the 25 μM sample were excluded due to excessive noise at <205 nm.” Please note that the spectra had excellent signal at 222 nm, which we used for analysis of helical content.
Although in the case of the LDH protein, the cryoprotective effect of YSK2 WT and constructs is comparable (PD50s are similar), most likely due to the equal radii of the various constructs, it is not clear why, in the case of the mutated constructs, the LDH activity is not only restored but also increased. Please comment this point.
This is an excellent point that we are not able to fully address at this time. We do not know why LDH activity is increased beyond 100% relative to the untreated control, and I have seen activities above 100% that vary from LDH batch to batch. Why the constructs appear to be even better will require more experiments.
Please discuss the possible reasons of the increase in the alpha helix content in Yfh1 at 1°C induced by YSK2 ET and KR compared to the native form at 25 ° C.
We think we have addressed this in the Discussion on lines 425-428, but would gladly provide more clarification if necessary.
Furthermore, I checked the sequences of YSK2 WT and mutants and I noticed that the socalled WT shows mutation H63N (Q9M605, uniport). Indicate whether this is an error in the sequence reported or whether this mutation actually exists. In the latter case, please report it in Materials and Methods.
This is an difference in the sequence reported for Q9M605 versus our sequence (i.e. we are not sure which one would be considered the mutation). For clarity, we have added the following text on Line 140: “Note that the sequenced used here contains a H63N substitution relative to the UniProt sequence Q9M605.”
If possible, also report the 15N-HSQC NMR spectra of YSK2 RK construct alone and in the presence of Yfh1.
Unfortunately we do not have chemical shift assignments for YSK2 R->K at this time.